# Preliminary Study on the Protective Effects and Molecular Mechanism of Procyanidins against PFOS-Induced Glucose-Stimulated Insulin Secretion Impairment in INS-1 Cells

**DOI:** 10.3390/toxics11020174

**Published:** 2023-02-14

**Authors:** Hai-Ming Xu, Meng-Yu Wu, Xin-Chen Shi, Ke-Liang Liu, Ying-Chi Zhang, Yin-Feng Zhang, Hong-Mei Li

**Affiliations:** 1School of Public Health and Management, Ningxia Medical University, Yinchuan 750004, China; 2The Key Laboratory of Environmental Factors and Chronic Disease Control of Ningxia, No. 1160, Shengli Street, Xingqing District, Yinchuan 750004, China; 3Institute for Translational Medicine, The Affiliated Hospital of Qingdao University, College of Medicine, Qingdao University, Qingdao 266021, China; 4The Key Laboratory of Fertility Preservation and Maintenance of Ministry of Education, Ningxia Medical University, Yinchuan 750004, China

**Keywords:** PFOS, glucose-stimulated insulin secretion (GSIS), insulin level, procyanidins, reactive oxygen species, INS-1 cells

## Abstract

This study aimed to investigate the effects of perfluorooctanesulfonic acid (PFOS) exposure on glucose-stimulated insulin secretion (GSIS) of rat insulinoma (INS-1) cells and the potential protective effects of procyanidins (PC). The effects of PFOS and/or PC on GSIS of INS-1 cells were investigated after 48 h of exposure (protein level: insulin; gene level: glucose transporter 2 (Glut2), glucokinase (Gck), and insulin). Subsequently, the effects of exposure on the intracellular reactive oxygen species (ROS) activity were measured. Compared to the control group, PFOS exposure (12.5, 25, and 50 μM) for 48 h had no significant effect on the viability of INS-1 cells. PFOS exposure (50 μM) could reduce the level of insulin secretion and reduce the relative mRNA expression levels of Glut2, Gck, and insulin. It is worth noting that PC could partially reverse the damaging effect caused by PFOS. Significantly, there was an increase in ROS after exposure to PFOS and a decline after PC intervention. PFOS could affect the normal physiological function of GSIS in INS-1 cells. PC, a plant natural product, could effectively alleviate the damage caused by PFOS by inhibiting ROS activity.

## 1. Introduction

PFOS is an important perfluorinated surfactant. Since the 1950s, due to its excellent stability, extremely low surface tension, and hydrophobic and oleophobic characteristics, it is widely used in various industries, such as firefighting agents, cosmetics, textiles, herbicides, floor polishing agent paint, kitchenware surface covers, and the production of food packaging [1,2]. In 2009, PFOS was listed in List B of the Stockholm Convention, which stipulated that the use of PFOS should be limited by the end of 2020 [3]. However, PFOS still exists in various environmental media because of its widespread use in the past decades and thus potentially exerts harmful effects on biota and human beings [4,5].

Recently, the role of PFOS in metabolism-related diseases, such as diabetes and hypertension, has attracted more and more attention [6]. The results of animal experiments showed that exposure to PFOS during pregnancy significantly affected the blood glucose level [7]. Using cellular assays, Duan et al. showed that insulin secretion was impaired in pancreatic β-cells under continual exposure to PFOS [8]. Epidemiological studies have yielded similar findings. For example, He et al. reported that serum PFOS exposure was positively associated with increased prevalence of diabetes in men [9]. Liu et al. systematically studied the association between total serum isomers of perfluorinated chemicals, glucose homeostasis, lipid profiles, serum protein, and metabolic syndrome in adults (National Health and Nutrition Examination Survey, 2013–2014). The results showed that serum isomers of PFOS were associated with glucose homeostasis, serum protein, and lipid profiles [10]. Based on the results of bibliometric analysis, it is fair to say that although some valuable scientific discoveries have been made, the molecular mechanisms of PFOS controlling the homeostasis of insulin metabolism and maintaining blood glucose balance are far from clear.

Insulin is a hormone secreted by the β cells of the pancreas. It is stimulated by various endogenous or exogenous substances and is the only hormone in the body that lowers blood sugar. In the body, insulin release is regulated under a very complex molecular network, including various endogenous and exogenous factors. Among these factors, glucose is the most important factor stimulating insulin secretion. In short, the principle of glucose-stimulated insulin secretion (GSIS) test is as follows. Glucose can stimulate islet β cells to increase insulin release, which can reflect the functional state of islet β cells, so it has certain value for the diagnosis, classification, and guidance of treatment of diabetes. For a long time, we have known that GSIS plays an important role in controlling the homeostasis of insulin metabolism and maintaining blood glucose within normal levels. However, we have just recently learnt that GSIS abnormalities can cause apoptosis in pancreatic β cells, which can then lead to functional decline and eventually induce noninsulin-dependent diabetes mellitus (NIDDM) [11,12].

Reactive oxygen species (ROS), as a signal molecule, participates in the regulation of insulin secretion [13]. Plant natural products are the treasure house of ROS scavengers. Therefore, many researchers have tried to screen ROS scavengers from plant natural products to cope with the inhibition of exogenous factors, such as environmental pollutants, on insulin secretion through the ROS pathway. Polyphenols are a group of phytochemicals with potential health-promoting effects (against the toxic effects of environmental pollutants). Polyphenols protect against chronic pathologies by modulating numerous physiological processes, such as cellular redox potential, enzymatic activity, cell proliferation, and signaling transduction pathways [14]. Procyanidins (PC) is a general term for a large group of polyphenolic compounds widely found in plants, including in the flowers, leaves, skins, shells, kernels, and seeds, with grape seeds containing the highest levels. An increasing number of trials have shown a correlation between adequate polyphenol consumption and a reduction in risk factors for chronic diseases, although deficiencies in polyphenol intake did not result in specific deficiency diseases [15,16]. Oboh et al. showed that the polyphenol extracts of jute leaf (*Corchorus olitorius*) could downregulate blood glucose levels by inhibiting α-amylase and glucosidase activities [17]. In addition, these compounds are promising in the treatment of chronic metabolic diseases, such as cancer, diabetes, and cardiovascular disease, as they prevent cell damage related to oxidative stress [18].

In conclusion, it is of great scientific and social significance to study whether perturbed insulin secretion and glucose metabolism by typical environmental pollutants can be antagonized or ameliorated by plant natural products. In this study, using rat insulinoma β-cells (INS-1) as in vitro cell model, the damage caused by PFOS exposure to GSIS function and the potential protective effect and preliminary molecular mechanism of phytochemical PC on PFOS-induced GSIS impairment were investigated.

## 2. Materials and Methods

### 2.1. Materials, Reagents and Instruments

The main experimental materials, reagents and instruments used in this study are shown in Appendix A.

### 2.2. Methods

#### 2.2.1. Cell Culture

INS-1 cells were cultured in RPMI-1640 medium containing 10% FBS, 1% penicillin–streptomycin solution, 5.6 mM glucose, 1 mM sodium pyruvate, and 50 μM β-mercaptoethanol. The cells were routinely cultured at 37.0 °C in an incubator containing 5% CO2 and saturated humidity. The medium was changed once every 2 days, and trypsin was used for passage when 80% confluence was reached.

#### 2.2.2. Preparation of PFOS and PC Solution

PFOS and PC powders were weighed and dissolved in DMSO. PFOS and PC solutions were filtered and sterilized by a 0.22 μm filter. The concentration required for subsequent experiments was prepared by gradient dilution.

#### 2.2.3. Cell Counting Kit-8 (CCK8) Assay

The cells in the logarithmic growth phase with good growth conditions were added to the culture medium to obtain uniform cell suspension and seeded into a 96-well plate at the density of 5 × 10^3^ cells/well. After 24 h of culture, the medium was aspirated and washed with PBS. The fresh complete medium containing a certain volume of PFOS solution was added into the wells of the experimental group to obtain final PFOS exposure concentrations of 12.5, 25, 50, 100, and 200 μM. The wells of the solvent control group were added with an equal volume of DMSO (0.1%) (n = 6). After 48 h of exposure, the cell viability was measured by the CCK8 kit. According to experience from previous experiments, the following two aspects were paid special attention to when carrying out the CCK8 experiment. First, considering the edge effect of the 96-well plate, the wells of the outermost circle were not inoculated with cells, and equal volume of PBS solution was added. Second, before and after adding CCK8 reagent, the cell density was examined by a microscope to confirm the consistency between different wells. 

According to the results of the CCK8 test, the doses that had no significant effect on cell viability were selected as the exposure doses of the subsequent experiments.

Using the same experimental method (the initial exposure doses of PC were 12.5, 25, and 50 μM, respectively), the exposure doses with no significant effect on cell viability were determined and selected as the exposure doses of subsequent experiments.

#### 2.2.4. Exposure Experiment

INS-1 cells in logarithmic growth phase with good growth condition were inoculated into a 24-well culture plate (2 × 10^5^ cells/well) and cultured for 24 h. Subsequently, for the alone exposure group, the cells were exposed to different doses of PFOS (12.5, 25, and 50 μM) and PC (6.25, 12.5, and 25 μM) for 48 h. For the combined exposure group, PFOS (50 μM) was pre-exposed for 6 h, and the medium was then aspirated and washed with PBS three times. Next, the cells were exposed to different concentrations of PC (6.25, 12.5, and 25 μM) for 6, 18, and 42 h. Finally, the medium was aspirated and washed with PBS three times for downstream experiments.

#### 2.2.5. ELISA Assay

After exposure, the supernatant was collected to determine the level of insulin secretion by ELISA with commercial kits according to the product instructions. Before formal determination, a reasonable dilution ratio was determined according to the preliminary experimental results to ensure that the determination results were within the linear determination range of the kits. According to the operational instructions of the corresponding kits, the OD values of the standards and samples were measured at 450 nm with a microplate reader. At the same time, the protein concentration in the supernatant was determined by the BCA method, which was used to standardize the data of insulin secretion level.

#### 2.2.6. Western Blot

After exposure, the cells were washed with ice-cold PBS and mixed with a RIPA lysis buffer to obtain lysates. Total protein in the supernatant was measured using a BCA protein assay kit. The proteins in the lysates taken from each sample were separated by SDS-PAGE, transferred to polyvinylidene difluoride membrane, blocked with 5% nonfat milk for 1 h at room temperature, and placed overnight at 4 °C with the primary antibodies insulin and β-actin with a dilution ratio of 1:1000. Then, the PVDF membrane was incubated with the corresponding secondary antibodies with a dilution ratio of 1:2000 at room temperature for 1 h. TBST was then used to wash the PVDF membrane three times. The immune complexes were detected with an enhanced chemiluminescence detection system. Densitometric analysis of the bands was performed with the ImageJ software. The relative protein expression level was calculated according to the ratio of the target protein level and the internal reference protein level.

#### 2.2.7. Real-Time Fluorescence Quantitative PCR (RT-qPCR)

The total RNA of each sample was extracted by the RNAsimple Total RNA Kit. The quality and integrity of the extracted total RNA was detected by an ultramicro ultraviolet spectrophotometer. Then, the total RNA was reverse transcribed by the FastKing gDNA Dispelling RT SuperMix Kit. The PCR reaction mixture was prepared according to the Real Universal Color Premix Kit instructions. The mRNA expression levels of *Glut2, Gck*, and insulin were detected by RT-qPCR. *Beta-actin* (β-actin) was used as the housekeeper gene. The primer sequences are shown in Appendix A. Three parallel wells were set for each sample. When the PCR amplification was completed, the relative mRNA expression levels were calculated by the classical 2^−ΔΔCT^ method [19].

#### 2.2.8. ROS Determination

INS-1 cells in the logarithmic growth phase with good growth conditions were inoculated into a 24-well culture plate (2 × 10^5^ cells/well) and cultured for 24 h. Subsequently, the cells were exposed to PFOS and PC alone or in combination in different concentration groups for 12, 24, and 48 h with or without glucose stimulation. After exposure, the medium was aspirated and washed with PBS three times for the determination of ROS level. Briefly, the probe was loaded in situ, and dichlorofluorescein diacetate (DCFH-DA) was diluted 1:1000 with a serum-free medium (500 μL/well). The cells were incubated in the incubator at 37 °C for 30 min. The cells were collected after washing with serum-free cell culture solution three times. Finally, the absorbance values were measured on a multidetection microplate reader at 480 (excitation wavelength) and 530 (emission wavelength) nm.

### 2.3. Statistical Analysis

SPSS 22.0 statistical software was used for statistical analysis. The experimental data were expressed as mean ± standard deviation. The relative expression levels of genes and proteins were compared by one-way ANOVA, and the pairwise comparison within the group was performed by the LSD method. The test level was α = 0.05, drawn by Graphpad 7.0 software.

## 3. Results

### 3.1. Effects of PFOS and PC after 48 h Exposure on the Viability of INS-1 Cells

The results of the CCK8 assay showed that the viability of INS-1 cells gradually decreased with increasing PFOS concentration at 48 h. Under the treatment of 100 and 200 μM PFOS, the viability of INS-1 cells significantly decreased by 23.16% and 38.81%, respectively (*p* < 0.05) (Figure 1a).

Under the treatment of PC for 48 h, the results of CCK8 assays showed that the viability of INS-1 cells increased by 3.4% and 1.4% at 6.25 and 12.5 μM PC, respectively (but the results were not statistically significant), and then decreased by 9.1% and 25.8% at 25 and 50 μM PC, respectively. Compared to the control group, the cell viability decreased significantly after exposure to 50 μM PC for 48 h (*p* < 0.05) (Figure 1b).

In this study, more attention was paid to the toxicological effects (PFOS) or protective effects (PC) caused by exposure doses that had no significant effects on cell viability. Therefore, in the subsequent exposure experiments, PFOS at 12.5, 25, and 50 μM and PC at 6.25, 12.5, and 25 μM were selected as the exposure doses.

### 3.2. Effects of PFOS and PC Alone or in Combination after 48 h Exposure on Insulin Secretion of INS-1 Cells under Glucose Stimulation

As can be seen in Figure 2a, the level of insulin secretion of INS-1 cells decreased by 32.2%, 36.3%, and 43.4% under 12.5, 25, and 50 μM PFOS treatment, respectively. Compared to the control group, the level of insulin secretion decreased significantly after exposure to 50 μM PFOS for 48 h (*p* < 0.05).

Under the treatment of PC, the results of ELISA showed that the level of insulin secretion of INS-1 cells increased gradually by 8.6%, 23.0%, and 33.3%. Compared to the control group, the level of insulin secretion increased significantly after exposure to 50 μM PC for 48 h (*p* < 0.05) (Figure 2b).

When the concentration of PC in the combined exposure group was 6.25 and 12.5 μM, the results of ELISA showed there was a significant difference compared to the control group (*p* < 0.05). Under different doses of PC and PFOS combined exposure, the level of insulin secretion of INS-1 cells increased gradually by 7.8%, 15.9%, and 26.3% compared to the PFOS group. When the concentration of PC reached 25 μM, the level of insulin secretion of INS-1 cells increased significantly (*p* < 0.05) (Figure 2c).

### 3.3. Effects of PFOS and PC Alone or in Combination after 48 h Exposure on the Level of Insulin Protein of INS-1 Cells Stimulated by Glucose 

As shown in Figure 3A, the relative expression of insulin protein in INS-1 cells decreased under the treatment of PFOS. Compared to the control group, the relative expression of insulin protein decreased significantly by 50% after exposure to 50 μM PFOS for 48 h (*p* < 0.05).

The relative expression of insulin protein in the PC group decreased compared to the control group (but the results were not statistically significant). Compared to the PFOS group, the relative expression of insulin protein in the combined exposure group increased significantly by 14%, 10%, and 12% when the concentration of PC in the combined exposure group was 6.25, 25, and 50 μM, respectively (*p* < 0.05) (Figure 3B).

### 3.4. Relative mRNA Expression Levels of Glut2, Gck, and Insulin in INS-1 Cells Exposed to PFOS and PC Alone or in Combination after 48 h Exposure under Glucose Stimulation

Figure 4a–c shows the relative expression of mRNA of *Glut2*, *Gck*, and insulin in INS-1 cells under the treatment of 12.5, 25, and 50 μM PFOS. Compared to the control group, the relative expression of mRNA of *Glut2*, *Gck*, and insulin decreased significantly by 51.9%, 37.9%, and 53.8%, respectively, after exposure to 50 μM PFOS for 48 h (*p* < 0.05).

Under the treatment of 6.25, 12.5, and 25 μM PC, compared to the control group, the relative expression of mRNA of *Glut2* and *Gck* increased significantly by 100% and 58%, respectively, after exposure to 25 μM PC for 48 h (*p* < 0.05). The relative expression of mRNA of insulin increased significantly by 91% and 76% after exposure to 12.5 and 25 μM PC for 48 h (*p* < 0.05) (Figure 4d–f).

The results of RT-qPCR showed that under the condition of different doses of PC and 50 μM PFOS simultaneously, the relative expression of mRNA of intracellular factors gradually increased compared to the PFOS group. When the concentration of PC reached 25 μM, the mRNA levels of *Glut2* and *Gck* increased significantly by 39.9% and 18.9%, respectively (*p* < 0.05). When the concentration was 12.5 μM and 25 μM, the mRNA level of insulin also increased significantly by 25.8% and 29.8% (*p* < 0.05) (Figure 4g–i).

### 3.5. ROS Levels in INS-1 Cells Exposed to PFOS and/or PC for 12, 24, and 48 h with or without Glucose Stimulation

At all timepoints, ROS levels were higher with glucose stimulation than without stimulation. PFOS exposure alone showed a trend of increasing and then decreasing ROS levels with or without glucose stimulation. When the cells were not stimulated by glucose, PFOS increased by 76.3% at 12 h, 223% at 24 h, and 53.5% at 48 h. When the cells were stimulated by glucose, PFOS increased by 53.3% at 12 h, 223% at 24 h, and 53.5% at 48 h. Exposure to PC reduced intracellular ROS levels with or without glucose stimulation (*p* < 0.05) (Figure 5).

## 4. Discussion

Recently, the role of PFOS in metabolism-related diseases, such as diabetes and hypertension, is attracting more and more attention [6]. After years of relevant scientific research, some valuable scientific discoveries have been made, but the molecular mechanisms of PFOS controlling the homeostasis of insulin metabolism and maintaining blood glucose balance are far from clear. 

In this study, we first focused on the influence of several key molecules that play an important role in regulating insulin secretion and maintaining the homeostasis level. Insulinoma β-cells express glucose transporter-2 (Glut2), which can be used as a sensor molecule to regulate insulin secretion. Glut2 imbalance is one of the early signs of NIDDM [20]. Glucokinase (GCK) is a key regulatory enzyme in insulinoma β-cells, which can phosphorylate glucose to glucose-6-phosphate. It plays a crucial role in the physiological process of regulating insulin secretion and is called a glucose sensor in insulinoma β-cells [21]. Of course, in addition, there are many related regulatory factors involved. On the whole, these regulatory factors constitute a complex, rigorous, and orderly regulatory network to jointly regulate insulin secretion of the body. In this study, we found that exposure to PFOS could downregulate the insulin secretion level of INS-1 cells stimulated by glucose and also affect the relative mRNA expression levels of Glut2, Gck, and insulin and the insulin protein level, suggesting that exposure to PFOS could affect the function of GSIS in INS-1 cells, thus leading to glucose metabolism disorder. Similarly, Wan et al. found that exposure to PFOS during pregnancy in CD-1 mice could even affect the disorder of glucose metabolism in their offspring (i.e., intergenerational toxic effects) [22].

PC is a new category of efficient natural antioxidants with strong in vivo viability and multiple functions of protecting the human body. Therefore, we studied whether PC could antagonize the functional damage of GSIS induced by PFOS and its preliminary molecular mechanism. Fortunately, some valuable results have been obtained. On the one hand, when INS-1 cells were exposed to PC alone, compared to the control group, the level of intracellular insulin secretion increased gradually with increasing PC exposure concentration. At the same time, the relative expression of Glut2, Gck, and insulin mRNA increased significantly (*p* < 0.05), but the expression level of insulin protein did not change significantly. On the other hand, when INS-1 cells were exposed to PC and PFOS, compared to the PFOS group, with the increase in PC concentration, the intracellular insulin secretion level; the relative expression of Glut2, Gck, and insulin mRNA; and the expression level of insulin protein increased gradually (*p* < 0.05). These results suggest that PC could antagonize the disturbance of intracellular insulin secretion and glucose metabolism caused by PFOS to a certain extent.

It is important to clarify the toxic effects caused by environmental pollutants, and it is more important to clarify the toxic mechanism. Based on the bibliometric analysis results and previous research results of our research group, we focused our attention on ROS, a sentinel molecule. ROS is a kind of oxygen-containing substance with active chemical properties and strong oxidation capacity in organisms. It includes superoxide anion radical (O_2_^−^), singlet excited oxygen (^1^O_2_), hydroxyl radical (·OH), hydrogen peroxide (H_2_O_2_), and peroxide (ROOH). ROS is very sensitive to environmental pollutants, and it is faster and more direct than the antioxidant defense system. It is an early warning indicator of environmental pollution. Studies have shown that some toxic effects caused by PFOS are mediated by ROS. For example, Zeng et al. found that PFOS increased intracellular ROS generation in L-02 cells in a concentration-dependent manner and that excessive ROS induced the reactive toxicity of cells, which eventually invoked autophagy [23]. Similarly, Qin et al. studied the impacts of PFOS on cell viability and insulin release capacity of pancreatic β cells using in vivo and in vitro methods. The results showed that the upregulation of ROS level caused by PFOS exposure mediated the above toxic effects [24]. In this study, we found that the damaging effect induced by PFOS on GSIS in INS-1 cells was also caused by the increase in ROS level. More importantly, the plant extract PC, an efficient ROS scavenger, could antagonize the above damaging effects at relatively low doses. In fact, in addition to ROS-mediated toxicity, PFOS can also produce toxicity through a variety of other direct or indirect ways. Our research group used a docking simulation method to study the possibility of direct interaction between PFOS and the proteins encoded by GCK, GLUT2, and INSR (insulin receptor) genes. Molecular docking results showed that PFOS could bind to the protein molecule encoded by GCK (Appendix A). It may also be one of the possible ways of GSIS impairment caused by PFOS exposure. Of course, in vivo and in vitro experiments are needed to verify the preliminary conclusions of this theoretical calculation. In view of the increasing evidence that PFOS has extensive and complex toxic effects on the human body, from the perspective of green environmental chemistry, developing new, environmentally friendly alternatives is recommended [25].

## 5. Conclusions and Perspectives

Taken together, PFOS could affect the normal physiological function of GSIS in INS-1 cells. PC, a plant natural product, could effectively antagonize the GSIS impairment caused by PFOS to a certain degree by inhibiting ROS activity. Next, we should pay close attention to the effects of different types of environmental pollutants (such as new persistent organic pollutants, nanomaterials, pharmaceuticals and personal care products, micro plastics, etc.) on the GSIS of in vitro and in vivo experimental models. Meanwhile, we should investigate the potential antagonistic effects of different types of phytochemicals on the functional damage of GSIS mediated by environmental pollutants. In view of the large variety of environmental pollutants and phytochemicals, it is recommended to use a high-throughput screening method to systematically carry out this research. We should focus on the impact of single or combined exposure of environmental pollutants with environmental related doses on GSIS, and simulate the exposure situation of the real environment as much as possible. Based on the 3R principle, more alternative toxicological methods should be employed to carry out relevant research rather than using a large number of experimental animals.

## Figures and Tables

**Figure 1 toxics-11-00174-f001:**
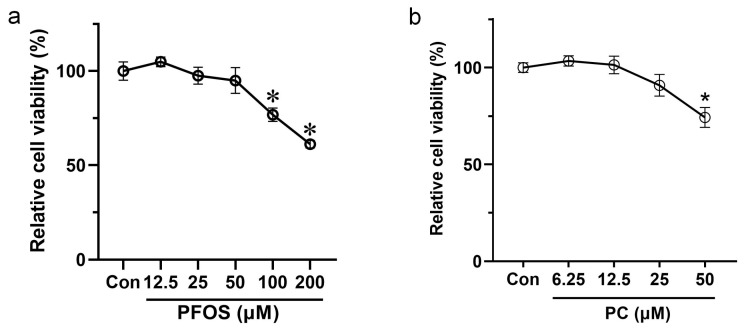
Effects of PFOS and PC on the viability of INS-1 cells. Values are expressed as mean ± SD, n = 6. (**a**) PFOS, (**b**) PC. * *p* < 0.05 vs. control.

**Figure 2 toxics-11-00174-f002:**
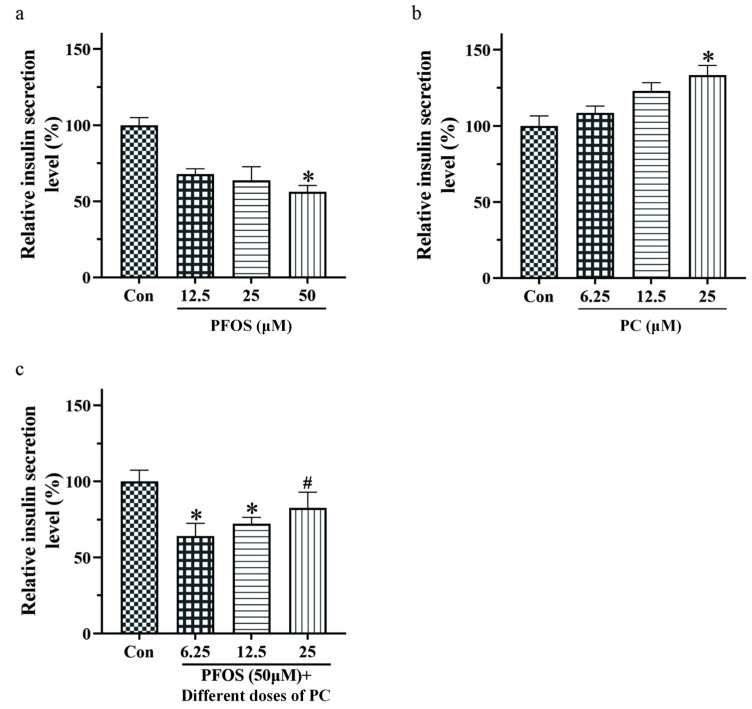
Effects of different doses of PFOS and PC after 48 h exposure on insulin secretion of INS-1 cells as determined by ELISA assay. Values are expressed as mean ± SD, n = 5, * *p* < 0.05 vs. control, # *p* < 0.05 vs. 50 μM PFOS. (**a**) PFOS; (**b**) PC; (**c**) 6.25, 12.5, and 25 μM PC and 50 μM PFOS in combination.

**Figure 3 toxics-11-00174-f003:**
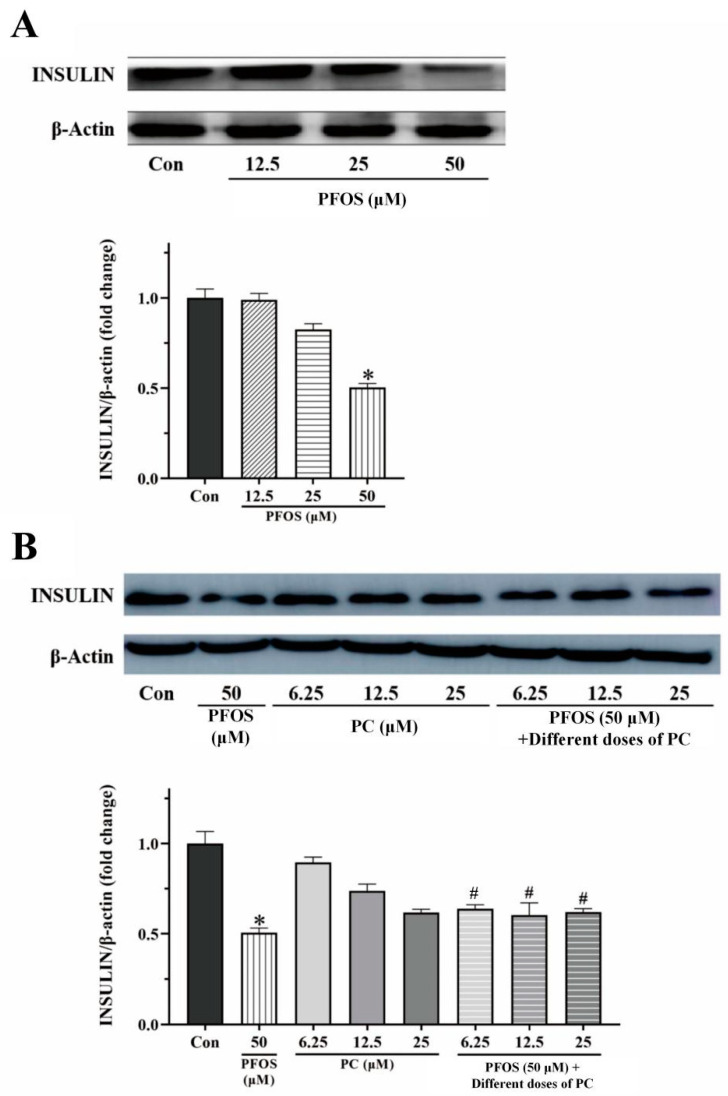
Effects of different doses of PFOS and PC after 48 h exposure on the relative expression of insulin protein in INS-1 cells as determined by Western Blot. Values are expressed as mean ± SD, n = 3, * *p* < 0.05 vs. control, # *p* < 0.05 vs. 50 μM PFOS. (**A**) PFOS, (**B**) PC and 50 μM PFOS alone or in combination.

**Figure 4 toxics-11-00174-f004:**
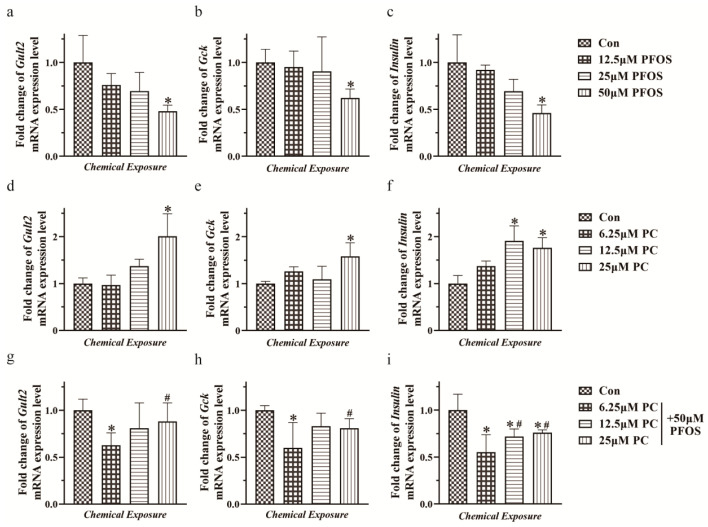
Effects of different doses of PFOS and PC after 48 h exposure on relative mRNA expression of Glut2, Gck, and insulin in INS-1 cells as measured by RT-qPCR assay. Values are expressed as mean ± SD, n = 3, * *p* < 0.05 vs. control, # *p* < 0.05 vs. 50 μM PFOS. (**a**–**c**) PFOS, (**d**–**f**) PC, (**g**–**i**) PC and 50 μM PFOS simultaneously.

**Figure 5 toxics-11-00174-f005:**
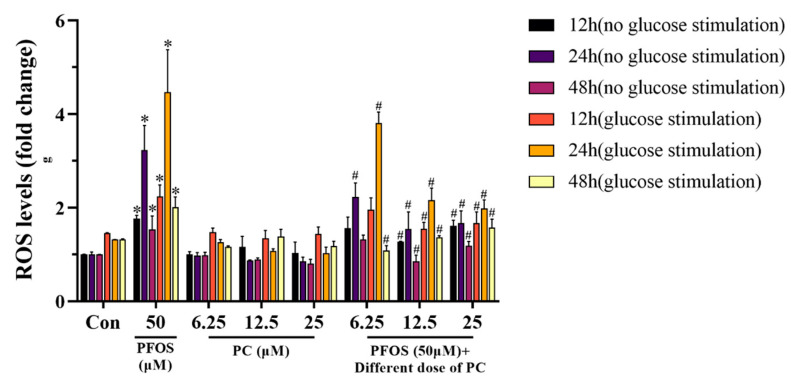
ROS levels in INS-1 cells exposed to PFOS and PC alone or in combination for 12, 24, and 48 h with or without glucose stimulation Values are expressed as mean ± SD, n = 3, * *p* < 0.05 vs. control. # *p* < 0.05 vs. 50 μM PFOS.

## Data Availability

All data generated or analyzed during this study are included in this published article.

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
