# Peer review of "Preliminary Study on the Protective Effects and Molecular Mechanism of Procyanidins against PFOS-Induced Glucose-Stimulated Insulin Secretion Impairment in INS-1 Cells"

_toxics, 2023, doi:10.3390/toxics11020174_

Round 1

Reviewer 1 Report

Overall well written paper with interesting results. I have minor comments below:

Line 27 and 30: ambiguous wording "to some extent" "certain degree",  specific % or fold changes should be mentioned as per reported in the results.

Line 74: maybe state a sentence or two on how polyphenols confer health-promoting effects.

Line 87: oddly worded sentence - do you mean to say "..to study how perturbed insulin secretion and glucose metabolism by typical.. can be antagonized or ameliorated by.."

Line 158: random primers

Line 162: parallel holes? do you mean wells?

Line 163: "classical" method, please provide citation.

Line 124 and 169: find better words than "sucked away", aspirated?

Line 114: what was the %DMSO used as solvent carrier for dosing?

Line 115: who was the vendor for the CCK8 kit?

Line 291: give full name of the acronym NIDDM

Line 366: Can the authors comment on the physiological relevance of their findings - as 50 uM is far above typical physiological ranges of PFOS (0.2 - 2 uM). As an alternative to animal systems, are in vitro assays able to reflect physiologically relevant adverse outcomes at realistic exposure concentrations?

Line 304 - 311: confusing paragraph, what do the authors mean by "node"
Author Response

Response Letter

Dear professor,

Thank you very much for reviewing our manuscript entitled "Preliminary study on the protective effects and molecular mechanism of procyanidins against PFOS-induced glucose stimulated insulin secretion impairment in INS-1 Cells" (Manuscript Number: toxics-2163418). We would like to thank you for your careful and thorough reading of this manuscript and for the thoughtful comments and constructive suggestions, which help to improve the quality of this manuscript. We expect that the submitted article will meet the journal's publication requirements and eventually be accepted.

Here, we try our best to reply as follows.

(1) Line 27 and 30: ambiguous wording "to some extent" "certain degree", specific % or fold changes should be mentioned as per reported in the results.

Response:

In our original version, there is indeed relevant information about this aspect. However, the instructions for authors require that the abstracts should be a total of about 200 words maximum. Therefore, in order to meet the requirements of the journal, we have made necessary reductions in the length of the abstract. Please refer to the results section for the details of the experimental results.

(2) Line 74: maybe state a sentence or two on how polyphenols confer health-promoting effects.

Response:

As reviewer suggested, "Polyphenols protects against chronic pathologies by modulating numerous physiolog-ical processes, such as cellular redox potential, enzymatic activity, cell proliferation and signaling transduction pathways." was added. Please refer to lines 80-82 of the revised manuscript.

(3) Line 87: oddly worded sentence - do you mean to say "..to study how perturbed insulin secretion and glucose metabolism by typical..can be antagonized or ameliorated by.."

Response:

It is really true as reviewer suggested that the statements of "In conclusion, it is of great scientific and social significance to pay attention to the disorder effects of insulin secretion and glucose metabolism caused by typical envi-ronmental pollutants and the potential antagonistic effect of plant natural products." was corrected as "In conclusion, it is of great scientific and social significance to study whether perturbed insulin secretion and glucose metabolism by typical environmental pollutants can be antagonized or ameliorated by plant natural products.". Please refer to lines 93-95 of the revised manuscript.

(4) Line 158: random primers

Response:

We are very sorry for our negligence. It has been modified. The statements of "The total RNA of each sample was extracted and cDNA was synthesized by reverse transcription using random primer." was corrected as "The total RNA of each sample was extracted by RNAsimple Total RNA Kit, the quality and integrity of the extracted total RNA was detected by ultra micro ultraviolet spectrophotometer. Then, the total RNA was reverse transcribed by FastKing gDNA Dispelling RT SuperMix Kit.". Please refer to lines 164-167 of the revised manuscript.

(5) Line 162: parallel holes? do you mean wells?

Response:

Yes, we are very sorry for our incorrect writing. The statements of "parallel holes" was corrected as "parallel wells". Please refer to line 171 of the revised manuscript.

(6) Line 163: "classical" method, please provide citation.

Response:

It has been provided citation. Please refer to references [19] of the revised manuscript.

(7) Line 124 and 169: find better words than "sucked away", aspirated?

Response:

It has been modified. The statements of "sucked away" was corrected as "aspirated" Please refer to lines 119 and 178 of the revised manuscript.

(8) Line 114: what was the %DMSO used as solvent carrier for dosing?

Response:

The statements of "0.1% DMSO" was added. Please refer to line 123 of the revised manuscript.

(9) Line 115: who was the vendor for the CCK8 kit?

Response:

The main experimental materials used in this study are shown in Supplemental files (Table S1).

(10) Line 291: give full name of the acronym NIDDM

Response:

We are very sorry for our negligence. It has been added to line 73, which is the first time NIDDM appears in the manuscript.

(11) Line 366: Can the authors comment on the physiological relevance of their findings - as 50 uM is far above typical physiological ranges of PFOS (0.2 - 2 uM). As an alternative to animal systems, are in vitro assays able to reflect physiologically relevant adverse outcomes at realistic exposure concentrations?

Response:

Thank the reviewer for this highly constructive comments. Here, we try to give feedback from the following two aspects: the typical environmental behavior characteristics of persistent organic pollutants (POPs) and the dose design of cytotoxicity experiments.

Long-term and low-dose exposure is one of the typical environmental behavior characteristics of POPs

POPs refer to natural or synthetic organic pollutants that are persistent, highly toxic, bioaccumulative, semi-volatile, and can migrate over a long distance through various environmental media and have serious harm to human health and the environment. POPs exposure, even at very low doses, may lead to various adverse health outcomes, including but not limited to metabolic diseases, nervous system diseases, immune system diseases, reproductive disorders, and interference with the normal development of infants and young children, which directly threaten human survival, reproduction and sustainable development.

Dose design of cytotoxicity experiment

In vitro cell exposure experiments are mostly short-term exposure. If the exposure dose is close to the internal exposure dose of natural people, short-term exposure is usually difficult to lead to measurable toxic effects. Therefore, at present, when designing the exposure dose of cytotoxicity experiments, researchers usually cover a relatively wide dose range in order to detect the toxic effects caused by environmental pollutants.

Of course, with the development of cytotoxicology (such as stem cell toxicology) and various high-throughput detection technologies (such as omics technologies), we can expect that in the near future, the exposure of environment-related doses will become the normal state of cytotoxicity research of environmental pollutants.

In addition, the limitations of this study and future research plans have been briefly described in the discussion and conclusion of the paper.

References

  • Hu L, Luo D, Wang L, Yu M, Zhao S, Wang Y, Mei S, Zhang G. Levels and profiles of persistent organic pollutants in breast milk in China and their potential health risks to breastfed infants: A review. Sci Total Environ. 2021 Jan 20;753:142028. doi: 10.1016/j.scitotenv.2020.142028. Epub 2020 Aug 29. PMID: 32906049.
  • Guillotin S, Delcourt N. Studying the Impact of Persistent Organic Pollutants Exposure on Human Health by Proteomic Analysis: A Systematic Review. Int J Mol Sci. 2022 Nov 17;23(22):14271. doi: 10.3390/ijms232214271. PMID: 36430748; PMCID: PMC9692675.
  • Yang R, Liu S, Yin N, Zhang Y, Faiola F. Tox21-Based Comparative Analyses for the Identification of Potential Toxic Effects of Environmental Pollutants. Environ Sci Technol. 2022 Oct 18;56(20):14668-14679. doi: 10.1021/acs.est.2c04467. Epub 2022 Sep 30. PMID: 36178254.
  • Liang X, Yang R, Yin N, Faiola F. Evaluation of the effects of low nanomolar bisphenol A-like compounds' levels on early human embryonic development and lipid metabolism with human embryonic stem cell in vitro differentiation models. J Hazard Mater. 2021 Apr 5;407:124387. doi: 10.1016/j.jhazmat.2020.124387. Epub 2020 Nov 2. PMID: 33172680.
  • Zhong G, Cui G, Yi X, Sun R, Zhang J. Insecticide cytotoxicology in China: Current status and challenges. Pestic Biochem Physiol. 2016 Sep;132:3-12. doi: 10.1016/j.pestbp.2016.05.001. Epub 2016 May 7. PMID: 27521907.

(12) Line 304 - 311: confusing paragraph, what do the authors mean by "node"

Response:

Considering the relevance of the content and for the purpose of simplification, we have deleted this paragraph in the revised manuscript.

The above is the detailed description of the revision of our manuscript. Thanks again for your hard work during the manuscript review process. We look forward to your reply.

Sincerely yours,

Dr. Hai-Ming Xu

Address: School of Public Health and Management, Ningxia Medical University, Yinchuan 750004, Ningxia, China

E-mail: xuhaiming5689467@163.com

Tel: 86-13469597910

Monday, January 30, 2023

Reviewer 2 Report

Manuscript ID: toxics-2163418

Summary:  This manuscript describes data from an in vitro experiment using pancreatic INS-1 cells exposed to PFOS, procyanidins, and a combination of PFOS and procyanidins.  The authors measured the effects of the test compounds and mixture on insulin excretion, insulin protein, gene expression, and reactive oxygen species.  There did appear to be an effect of procyanidins counteracting the PFOS effects, however the effects of PFOS on cell viability also appear to significantly confound the results.

Comments:

1.       Figure 1 – The data shows that even the lowest dose of PFOS tested reduced cell viability ~25%.  Even though this was not statistically significant from your analyses, it is a rather large reduction in cell viability.  Lower doses below 12.5 uM should have been tested to determine dose ranges that did not kill so many of the cells.

2.       Figure 2 – in relation to the above comment, the reduction in relative insulin secretion acros PFOS dose levels appears to simply mirror the effects on cell viability so as a reader I have little confidence that the reduction in insulin is a mechanistic effect of PFOS as opposed to an reduction just because fewer cells are alive.  This comment essentially applies to all of the data generated because the mixture exposure used a dose of 50uM PFOS and the cell viability data shows that dose killed ~30% or more of the cells.

3.       Line 42: This statement should be qualified to “and thus potentially exerts harmful effects on biota and human beings.”  Hazard is a function of dose and not all biota and human beings are exposed to harmful amounts of PFOS.

4.       Line 63: I think “release” should be changed to “secretion” so that the GSIS abbreviation makes sense.

5.       Line 69: NIDDM needs to be defined.

6.       Line 83: “amylase” is repeated

7.       Lines 103-106: As this is a toxicology study the test chemicals must clearly be reported.  For both PFOS and PC the exact CASRN, manufacturer, lot, and purity must be included here in the main body of the paper.

8.       Line 107: Define CCK8

9.       Line 110: Change “sucked away” to “aspirated”

10.   Lines 113-114: What was the concentration of DMSO vehicle? Was it consistent across all dose levels and vehicle controls?

11.   Lines 126-133: This paragraph needs to be edited for clarity.  It is not clear how the experiments were conducted.  Were the cells co-exposed to both PFOS and PC? I assume so but can’t tell what doses of either chemical were used.

12.   Lines 161-162: It’s not clear what meant by “Three parallel holes were set for each sample.” What is a hole?

Author Response

Response Letter

Dear professor,

Thank you very much for reviewing our manuscript entitled "Preliminary study on the protective effects and molecular mechanism of procyanidins against PFOS-induced glucose stimulated insulin secretion impairment in INS-1 Cells" (Manuscript Number: toxics-2163418). We would like to thank you for your careful and thorough reading of this manuscript and for the thoughtful comments and constructive suggestions, which help to improve the quality of this manuscript. We expect that the submitted article will meet the journal's publication requirements and eventually be accepted.

Here, we try our best to reply as follows.

Summary: This manuscript describes data from an in vitro experiment using pancreatic INS-1 cells exposed to PFOS, procyanidins, and a combination of PFOS and procyanidins. The authors measured the effects of the test compounds and mixture on insulin excretion, insulin protein, gene expression, and reactive oxygen species. There did appear to be an effect of procyanidins counteracting the PFOS effects, however the effects of PFOS on cell viability also appear to significantly confound the results.

  • Figure 1 – The data shows that even the lowest dose of PFOS tested reduced cell viability ~25%.Even though this was not statistically significant from your analyses, it is a rather large reduction in cell viability. Lower doses below 12.5 uM should have been tested to determine dose ranges that did not kill so many of the cells.

Response:

Thank the reviewer for this highly targeted and constructive comment. After receiving the revise notice, we traced the experimental results in the first submitted version and found two defects in the details of the experiment. First, the edge effect of 96-well plate was not fully considered. Second, there was no microscopic examination of cell density before and after adding CCK-8 reagent (although theoretically, cell density between different wells should be approximately equal). These two details have a direct impact on the experimental results, but are easy to be ignored.

Based on this, we supplemented the INS - cell viability experiment caused by PFOS. During the experiment, we paid special attention to the above two experimental details. More objective and realistic experimental data are obtained (as shown in the figure below).

Figure 1 Different doses exposed for 48 h on the viability of INS-1 cells measured by CCK8 assay. Values were expressed as mean ± SD, n=6. (a) PFOS, (b) PC.

(2) Figure 2 – in relation to the above comment, the reduction in relative insulin secretion acros PFOS dose levels appears to simply mirror the effects on cell viability so as a reader I have little confidence that the reduction in insulin is a mechanistic effect of PFOS as opposed to an reduction just because fewer cells are alive. This comment essentially applies to all of the data generated because the mixture exposure used a dose of 50uM PFOS and the cell viability data shows that dose killed ~30% or more of the cells.

Response:

In terms of scientific logic, this question is the same as the first one above. Please refer to our reply to the first question.

(3) Line 42: This statement should be qualified to “and thus potentially exerts harmful effects on biota and human beings.” Hazard is a function of dose and not all biota and human beings are exposed to harmful amounts of PFOS.

Response:

We have made correction according to the Reviewer's comments. The statements of "However, PFOS still exists in various environment media because of its widespread use in the past decades and thus exerts harmful effects on biota and human beings." was corrected as "However, PFOS still exists in various environment media because of its widespread use in the past decades and and thus potentially exerts harmful effects on biota and human beings.". Please refer to lines 42-44 of the revised manuscript.

(4) Line 63: I think “release” should be changed to “secretion” so that the GSIS abbreviation makes sense.

Response:

We are very sorry for our incorrect writing. The statements of "release" was corrected as "secretion". Please refer to line 66 of the revised manuscript.

(5) Line 69: NIDDM needs to be defined.

Response:

We are very sorry for our negligence. It has been added to lines 69-73.

(6) Line 83: “amylase” is repeated

Response:

"amylase" was deleted. We are very sorry for our negligence.

(7) lines 103-106: As this is a toxicology study the test chemicals must clearly be reported. For both PFOS and PC the exact CASRN, manufacturer, lot, and purity must be included here in the main body of the paper.

Response:

PFOS (Sigma-Aldrich Company, USA, CASRN: 1763-23-1, Lot: 33607, Purity: ≥99%) and PC (Melone Pharmaceutical Co., Ltd., China, CASRN: 4852-22-6, Lot: S0208A, Purity: ≥95%) were added. Please refer to lines 111-113 of the revised manuscript.

(8) Line 107: Define CCK8

Response:

Cell counting kit-8 (CCK8) assay was added. Please refer to line 116 of the revised manuscript.

(9) Line 110: Change “sucked away” to “aspirated”

Response:

We have corrected according to the reviewer's suggestion. Please refer to lines 119 and 178 of the revised manuscript.

(10) lines 113-114: What was the concentration of DMSO vehicle? Was it consistent across all dose levels and vehicle controls?

Response:

The statements of "0.1% DMSO" was added. DMSO was consistent across all dose levels and vehicle controls. Please refer to line 123 of the revised manuscript.

(11) lines 126-133: This paragraph needs to be edited for clarity. It is not clear how the experiments were conducted. Were the cells co-exposed to both PFOS and PC? I assume so but can’t tell what doses of either chemical were used.

Response:

We have re-written Part 2.2.4 Exposure experiment according to the reviewer's suggestion. As below: INS-1 cells in logarithmic growth phase with good growth condition were inoculated into 24-well culture plate (2×105 cells/well) and cultured for 24 h. Subsequently, for the alone exposure group the cells were exposed to different doses of PFOS (12.5, 25, 50 μM) and PC (6.25, 12.5, 25 μM) for 48 h. For the combined exposure group, PFOS (50 μM) was pre-exposed for 6 h and then medium was aspirated and washed with PBS three times. Next, the cells were exposed to different concentrations of PC (6.25, 12.5, 25 μM) for 6, 18, 42h. Finally, the medium was aspirated and washed with PBS three times for downstream experiments. Please refer to lines 131-138 of the revised manuscript. 

(12) lines 161-162: It’s not clear what meant by “Three parallel holes were set for each sample.” What is a hole?

Response:

We are very sorry for our incorrect writing. The statements of "parallel holes" was corrected as "parallel wells". Please refer to line 171 of the revised manuscript.

The above is the detailed description of the revision of our manuscript. Thanks again for your hard work during the manuscript review process. We look forward to your reply.

Sincerely yours,

Dr. Hai-Ming Xu

Address: School of Public Health and Management, Ningxia Medical University, Yinchuan 750004, Ningxia, China

E-mail: xuhaiming5689467@163.com

Tel: 86-13469597910

Monday, January 30, 2023

Round 2

Reviewer 2 Report

The authors addressed my prior comments.  However, I do have to say that it is a bit suspicious that the PFOS cell viability data changed dramatically with no change in the controls and no additional detail provided in the Methods.  Further, the PC cell viability data didn't change at all from the last time.  The author response isn't clear whether the viability assay was rerun differently, or the original data was just reanalyzed in a different way.
